# Shifting Epidemiology Trends in Tongue Cancer: A Retrospective Cohort Study

**DOI:** 10.3390/cancers15235680

**Published:** 2023-12-01

**Authors:** Yara Sakr, Omar Hamdy, Maher Eldeghedi, Rabab Abdelaziz, Echreiva Med Sidi El Moctar, Mohammed Alharazin, Shadi Awny

**Affiliations:** 1Faculty of Medicine, Mansoura University, Mansoura 35516, Egypt; sakryara96@gmail.com (Y.S.); mseldeghedi@outlook.com (M.E.); m0599449025ohammed@std.mans.edu.eg (M.A.); 2Surgical Oncology Department, Oncology Center, Mansoura University, Mansoura 35516, Egypt; shadiawny@mans.edu.eg; 3Faculty of Medicine, Cairo University, Cairo 11559, Egypt; rabababdelaziz683@gmail.com; 4Faculty of Medicine, Ain Shams University, Cairo 11591, Egypt; dr.echreiva@gmail.com

**Keywords:** tongue cancer, gender, young, epidemiology

## Abstract

**Simple Summary:**

Tongue cancer typically has male predominance. However, several studies have documented an increasing number of incidences among the younger population, with female predominance, which is unusual. Current trends in tongue cancer and possible risk factors require more investigation. The global epidemiological trends in tongue cancer have shifted toward more incidences in younger and female groups. Our study confirms the same findings regarding the younger age group. However, in our locality, no significant changes have been noted regarding gender. More multi-institutional studies are recommended to explain such changes and highlight the possible risk factors so that appropriate corrective actions can be taken into consideration.

**Abstract:**

The tongue is the most common site for oral cavity carcinoma. It typically has male predominance. However, several studies have documented an increasing number of incidences among the younger population, with female predominance, which is unusual. In this study, we aimed to determine current trends in tongue cancer regarding age and gender. Data from 197 tongue cancer patients were extracted from The Oncology Center, Mansoura University (OCMU) database from 2006 to 2021. The patients were divided into two time periods: (2006–2013) and (2014–2021). We computed counts and proportions of tongue cancer for demographic and tumor characteristics. The data were analyzed using SPSS. Gender showed no statistically significant difference in both groups, while the percentages of diagnosed females were 52.7% and 52%, respectively. The percentages of males were 47.3% and 48%, *p*-value = 0.927. There was a statistically significant difference in the number of patients aged 20 to 39 years old and ≥60 years old in both periods. The *p*-values were 0.039 and 0.011, respectively. Although tongue cancer is typically more common in males, our results showed no significant difference in the gender of diagnosed patients. In addition, our results showed that the number of younger patients significantly increased in the period from 2014 to 2021. However, we encourage further investigations involving larger populations.

## 1. Introduction

With over 300,000 new cases diagnosed and 177,000 deaths worldwide, oral cancer is considered a global health issue, especially in less developed countries lacking diagnosis and treatment facilities [1]. In Egypt, oral cancer ranks 21st in newly diagnosed cases of cancer and 20th in cancer-related deaths [2]. Tongue cancer is the commonest and most aggressive type of oral cavity malignancy [3,4]. Although its etiology is multifactorial, tobacco, alcohol, and betel quid are quintessential contributors, especially in Africa and the Middle East where smoking is a major problem [5,6,7]. In more recent years, the human papillomavirus (HPV) has been recognized to be an independent cause of oropharyngeal squamous cell carcinoma (SCC).

While the overall number of head and neck squamous cell carcinoma (HNSCC) incidences continues to diminish globally due to reduced cigarette and alcohol use, there have been reports of an increase in the number of incidences of oral tongue cancer in numerous nations. It is interesting to note that just a small percentage of oral squamous cell carcinomas appears to be HPV-related. As a result, an increasing number of HNSCC patients do not appear to have a long history of drinking or smoking. This may indicate the existence of risk factors other than alcohol and cigarettes [8].

Although tongue cancer is commonly seen in males over 50 years old [4], some researchers have claimed that there is an increasing incidence in the younger population [9,10]. In addition, in another study including 22 tumor registries, the results demonstrated a yearly increase ranging from 0.4% to 3.3% in tongue cancer incidence with a predominance in younger patients in some registries, while other registries showed almost equal distributions and even male predominance [10,11,12,13,14,15]. Nevertheless, few studies have investigated head and neck cancers in the younger, non-smoking population [12]. Even though tongue cancer is a global burden, there is not enough data on its epidemiology, and accurate statistics are lacking in the Middle East and North Africa (MENA) region including Egypt. Although some studies have investigated the prevalence of head and neck cancers in Egypt [16], current trends in tongue cancer and possible risk factors require more investigation, especially with the global rise in the number of tongue cancer incidences at younger ages. A study conducted from 1973 to 2012 showed an increase in the number of incidences of tongue cancer between men and women, in addition to several descriptive studies on trends in oral tongue cancer, indicating that the number of incidences has dramatically increased among young (ages 18–44), White people, mainly women [17].

Another global study, involving 80,000 people from a total of twenty-two countries, showed that the incidence of this disease as a global phenomenon, especially among people of a youthful age, with a particular focus on developing countries, which have a high prevalence of head and neck cancers [18].

It is essential to point out that trends in tongue carcinoma are debatable, and finding precise data for subgroups can be challenging. This motivated us to investigate changing trends in tongue cancer based on the Oncology Center Mansoura University (OCMU) database.

## 2. Materials and Methods

### 2.1. Source and Variables

In this retrospective study, cases diagnosed with tongue cancer between 2006 and 2021 were extracted from the Oncology Center Mansoura University (OCMU) database. Cases were divided according to the year of diagnosis into two groups: the old group (2006–2013) and the recent group (2014–2021). For this study, we extracted the following variables: age at diagnosis (years), gender, smoking status (smoker or non-smoker), family history (positive or negative), and type of cancer reported by patients in family history. Also, tumor characteristics were obtained, including mass side (right or left), site (lateral, dorsum, or middle/posterior), nodal status at presentation based on post-operative histology (positive or negative), clinical stage, and grade. Age was categorized into three groups: young adults (20–39 years old), adults (40–59 years old), and elderly (≥60 years old) to compare the number of diagnosed tongue cancer cases in each age category in both periods.


**Eligibility criteria**


Patients ≥18 years old;Patients with primary tongue cancer who were diagnosed with biopsy.


**Exclusion criteria**


Patients with recurrent or secondary tongue cancer;Patients diagnosed with tongue cancer before 2006 and after 2021.

### 2.2. Statistical Analysis

The statistical software program SPSS version 28.0 was used for the analysis of this study. Descriptive statistics for included patients were presented in the form of frequencies and percentages in the two time periods (2006–2013) and (2014–2021). We also summarized counts and proportions of tumor characteristics at diagnosis.

Univariate analysis was performed using the chi-square test for categorical variables. Mann–Whitney and Student’s *t*-tests were used based on the normality distribution test for continuous variables. The *p*-value was considered significant at a level of <0.05.

## 3. Results

Data from 197 tongue cancer patients were retrieved from the Oncology Center Mansoura University (OCMU) database. They were divided into two groups of time periods according to the year of diagnosis. Group one (older) refers to cases diagnosed between 2006 and 2013 while group two (recent) refers to cases diagnosed between 2014 and 2021. The patient characteristics included in the study are shown in Table 1. Group one (older) included 74 patients (37.6%) while group two (recent) included 123 patients (62.4%). The tumor characteristics at presentation are also summarized in Table 2.

### 3.1. Patents’ Characteristics


**Gender**


There was no statistically significant difference in gender between the older and recent groups. The percentages of females diagnosed with tongue cancer in both groups were 52.7% and 52%, respectively. Meanwhile, the percentages of diagnosed males were 47.3% and 48%, *p*-value = 0.927. Figure 1 demonstrates the numbers of males and females diagnosed with tongue cancer over the period from 2006 to 2021.


**Age at diagnosis (Figure 2)**

**Age from 20 to 39 years**


In the period of 2006–2013 (older), five (6.8%) patients diagnosed with tongue cancer were between 20 and 39 years old compared to seventeen patients (17.8%) in 2014–2021 (recent). The difference was statistically significant, indicating a significant increase in patients diagnosed with tongue cancer in this age category, *p*-value = 0.039. Figure 3A demonstrates the frequencies of tongue cancer cases aged from 20 to 39 years diagnosed in both periods, comparing their mean ranks. The lower mean rank in the recent group indicates that within this age category, patients diagnosed with tongue cancer were at younger ages compared to the same age category in the older group, which, accordingly, indicates an increase in tongue cancer cases among the younger population.


**Age from 40 to 59 years**


Our results showed no statistically significant difference in the numbers of diagnosed tongue cancer patients in both periods. In total, 22 (29.7%) were diagnosed in the older group compared to 53 (43.1%) in the recent group, *p*-value = 0.912. Figure 3B demonstrates frequencies of cases aged 40 to 59 years diagnosed in both periods, comparing their mean ranks. The mean ranks of both groups are relatively close, indicating close ages at diagnosis in both groups. Accordingly, there is no significant difference in the number of patients diagnosed with tongue cancer in this age category between the old and the recent groups.


**Age**
**≥60 years**


There was a statistically significant increase in the number of patients diagnosed with tongue cancer in this age group. A total of 47 (63.5%) patients were diagnosed in the older group, compared to 53 (43.1%) diagnosed in the recent group, *p*-value = 0.011. Figure 3C demonstrates frequencies of tongue cancer cases aged ≥60 years diagnosed in both periods, comparing their mean ranks. The older group shows a higher mean rank, which means that although there is a significant increase in tongue cancer cases among patients ≥60 years in the recent group, patients diagnosed with tongue cancer in the older group were relatively older than patients in the recent group with a lower mean rank.

The number of total population in each age group is shown in Figure 4.

**Figure 2 cancers-15-05680-f002:**
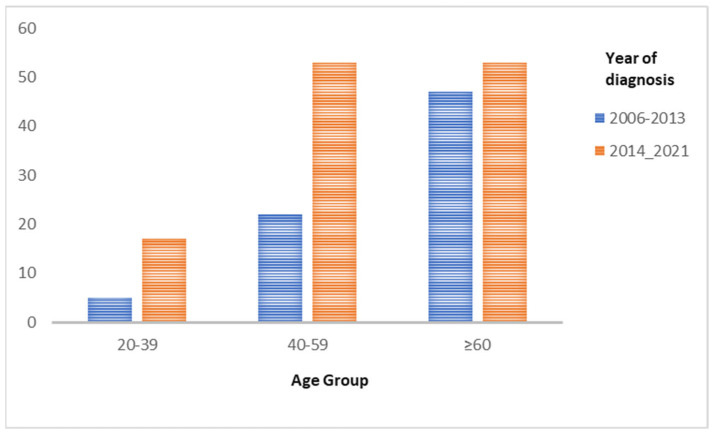
Bar charts demonstrating the distribution of patients’ age groups in two time periods (2006–2013) and (2014–2021).

**Figure 3 cancers-15-05680-f003:**
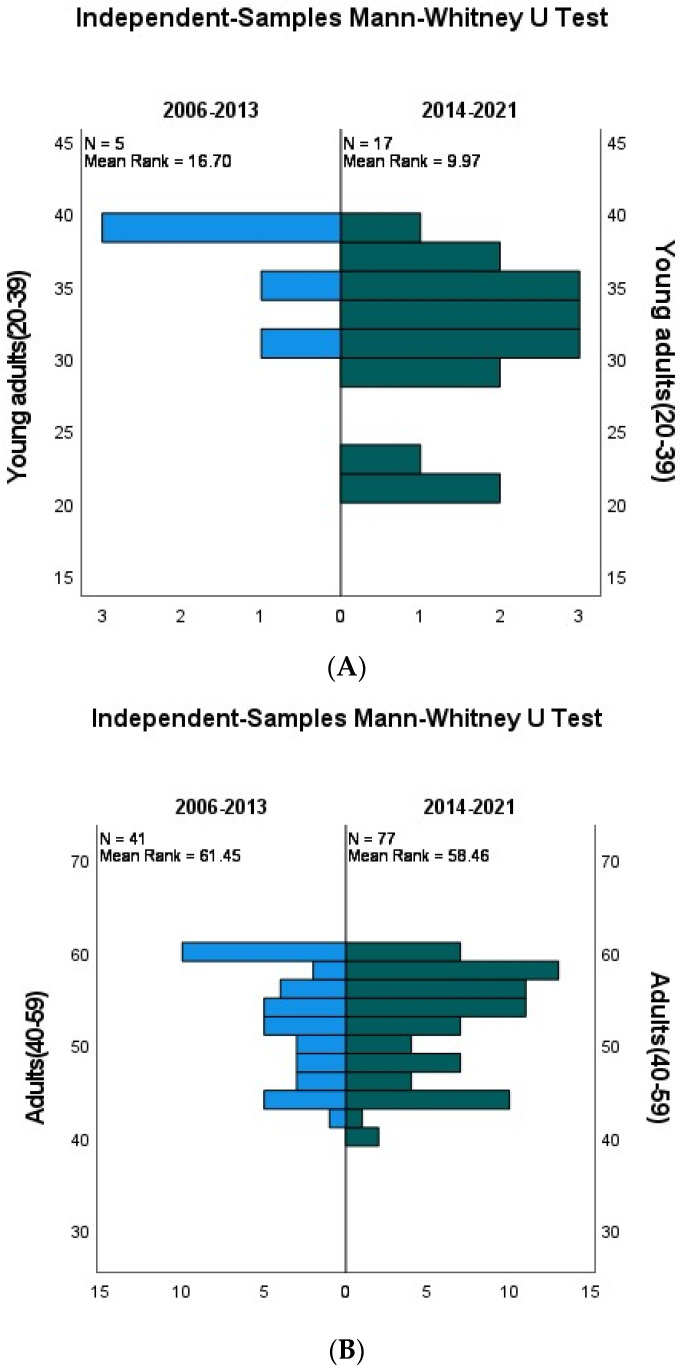
(**A**) Graph demonstrating the numbers of patients diagnosed with tongue cancer in the age group (20–39 years old) in two time periods with their mean ranks. (**B**) Graph demonstrating the numbers of patients diagnosed with tongue cancer in the age group (40–59 years old) in two time periods with their mean ranks. (**C**) Graph demonstrating the numbers of patients diagnosed with tongue cancer in the age group (more than 60 years old) in two time periods with their mean ranks.

**Figure 4 cancers-15-05680-f004:**
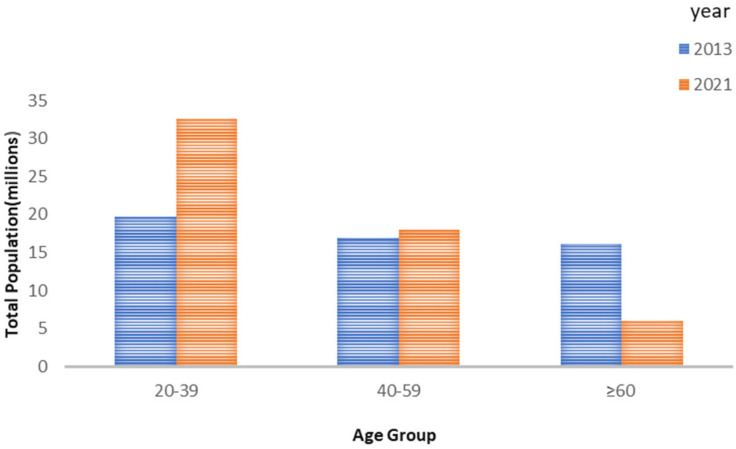
Bar chart showing numbers of total population in three age groups in 2013 and 2021.

### 3.2. Smoking Status

The numbers of smokers were 17 (23%) and 28 (22.8%) in both groups, respectively, while the numbers of non-smokers were 57 (77%) and 95 (77.2%), respectively. Smoking status was not statistically significant between the two groups, *p*-value = 0.973.

### 3.3. Family History

#### 3.3.1. Negative Family History

A total of 64 (98.5%) tongue cancer patients in the older group reported a negative family history of malignancy compared to 94 (83.2%) in the recent group.

#### 3.3.2. Positive Family History

One patient in the older group reported a family history of breast cancer. Meanwhile, in the recent group, 19 (16.8%) patients reported a positive family history of cancer, which included seven cases of head and neck cancers including thyroid, mandible, and larynx cancers, seven GIT cancer cases including four cases of hepatic carcinoma, and two cases of stomach cancer. In addition, nine gynecological malignancies including two breast and six uterine cancer cases were reported. Other malignancies such as prostate, brain stem, leukemia, and lung cancer were also reported. Family history was statistically significant, with *p*-value = 0.002.

### 3.4. Tumor Characteristics

Regarding tumor site, most patients presented with a mass on the lateral side of the tongue, with 57 (77%) patients in the older group compared to 103 (83.7%) in the recent group. Meanwhile, the least common site was the middle/posterior side, *p*-value = 0.253. A total of 46 (62.2%) patients in the older group presented with a positive nodal status compared to 41 (33.3%) in the recent group. Meanwhile, 26 (35.1%) in the older group presented with a negative nodal status compared with 55 (44.7%) in the recent group. Nodal status was statistically significant, *p*-value < 0.001. Most patients were presented in clinical stage IV, with 42 (60%) patients in the older group compared to 36 (43.4%) in the recent group, *p*-value = 0.148. The tumor characteristics at presentation are summarized in Table 2**.**

## 4. Discussion

Oral cancer is the 11th most common malignancy in the world. Apart from non-melanoma skin cancer, oral cancer is the most common head and neck malignancy, with the lip and tongue being the most involved subsites and squamous cell carcinoma being the most common histological type [19]. Extensive research has been conducted exploring the trends of tongue cancer. Interestingly, tongue cancer demonstrates significant geographic variations which might be due to cultural, habitual, or religious differences among different countries. Most of the available studies were conducted in Western countries with a significant paucity of data from other regions of the world, specifically Muslim communities. In addition, the multifactorial nature of the disease makes approaching the problem, analyzing risk factors, and monitoring trends trickier and highly variable, not only on the international level, but also on the national level. For example, a review article summarizing the global incidence of tongue cancer—with little data on Africa—demonstrates the variations in tongue cancer trends among different countries and different states within the US [20]. Therefore, in this study, we aim to demonstrate tongue cancer trends based on our data from the Oncology Center Mansoura University (OCMU) with a focus on squamous cell carcinoma and to lay the groundwork for further studies reflecting trends, specifically in Muslim countries.

According to our study, there was no significant difference in the prevalence of tongue cancer between males and females.

Several studies have been conducted to demonstrate the relationship between gender and tongue cancer. According to the available data, males showed higher rates of tongue cancer in some studies, females had higher rates in others, while some studies have claimed that gender is of little significance. However, variations can be explained by the differences in survival between males and females demonstrated in some studies, which are also debatable. Meanwhile, in some studies, females have been shown to have a better prognosis because of different smoking and alcohol habits compared to males and due to the fact that females tend to seek medical attention relatively earlier than males, which positively impacts prognosis with early detection and can lead to apparent increased prevalence among females [21,22]. However, other studies have claimed that men have better OS [23]. Data from the Netherlands cancer registry between 1989 and 2006 showed an increasing number of incidences of tongue SCC, with 2.1% in women and 1.1% in men [24]. In contrast, an analysis of the NORDCAN registry (Sweden, Norway, Finland, Denmark, and Iceland) from 1960 to 2008 found a greater increase in the age-standardized incidence rate of tongue SCC in men than in women [25]. Different habits between countries may play a role in the gender distribution of tongue cancer, for example, alcohol, which is a worldwide known risk factor for tongue cancer. Alcohol consumption is less in Arab Muslim countries due to the cultural and religious backgrounds [26]. This low intake of alcohol may show a statistical difference in gender distribution compared to non-Muslim countries. Unfortunately, we lack studies to support such a hypothesis. In addition, it is crucial to mention that drinking alcohol is less admitted by individuals in Muslim countries, especially by women [27]. However, we cannot admit that men have lower rates than in the West because alcohol is not the only risk factor. In all, we need more studies, especially in Muslim communities. Interestingly, differences in incidence rates between genders have been attributed to hormonal differences in some studies, where it was presumed that estrogen may have a vital role in increasing precancerous cell movement in the mouth and promoting the spread of head and neck cancers [13]. In a recent report, 75% of young never-smoker/never-drinker HNSCC patients who developed primary oral tongue squamous cell carcinoma were women, concluding that female hormones may have a significant role in carcinogenesis [28]. Recent studies have claimed that higher prolactin levels in HNC can be a marker of poor prognosis [29], and in another study, levels of prolactin, estrogen, progesterone, and DHEAS were measured in male patients with tongue cancer, and the results demonstrated a decreased ratio of testosterone:estradiol and increased levels of FSH, LH, and prolactin, which align with other studies claiming that prolactin might have a role in tongue cancer. Despite the general trend of male predominance [28,30], given the highly debatable association between gender and tongue cancer and the scarcity of valid statistics reflecting trends, specifically in Muslim countries, it is even more challenging to establish a clear trend in tongue cancer regarding gender.

Based on our results, age groups of 20–39 and over 60 years old demonstrated a significant increase in tongue cancer cases in the period of 2014–2021 compared to 2006–2013. Meanwhile, the age group of 40–59 years old showed no difference throughout the study.

Age is a crucial risk factor for cancers generally and tongue cancer specifically. Although it is more common for head-and-neck-cancer incidences to peak in older ages and with chronic use of tobacco and alcohol, there has been an increase in the number of tongue cancer incidences among the younger population, where around 5% of patients are diagnosed below 45 years of age, and, according to available data, this trend seems to be consistently increasing [31]. According to a global study including twenty-two registries, a yearly increase from 0.4% to 3.3% with significantly increased rates in younger patients has been demonstrated in fourteen out of twenty-two registries [32]. However, the results of the available studies are mainly heterogeneous because of the complex dynamics of the disease, variation in the reporting of cases, and the crucial cultural and habitual differences among countries, including smoking, alcohol, and HPV infection, which are crucial risk factors for tongue cancer. The increasing number of incidences of tongue cancer in the younger population has been well documented in several studies [32,33]. Studies are liable to selection bias and the heterogeneity of included patients, both of which must be taken into consideration. Nevertheless, other studies [34,35,36] have demonstrated the absence of significant differences in the younger population (<45–40 years old). However, several studies that used a matched analysis have reported an increase in relapses in young adults [37,38,39]. Most available studies have used a threshold of 40–45 years old to define the younger population. However, according to our observation at OCMU, there have been several cases documented in the twenties and early thirties, which encouraged us to further subdivide age categories for a better demonstration of tongue cancer cases in significantly younger populations. In addition, some studies have demonstrated significantly high rates of tongue cancer in 30-year-old patients and lower [18,38], which raises the question of how young can tongue cancer be diagnosed. A unique feature differentiating between tongue cancer in young and old people is the absence of significant risk factors such as tobacco and alcohol, as they require a more chronic use exposure to cause malignancy. In [40,41], it is suggested that smoking for 21 years may be necessary to increase the risk of oral cancer. This amount of exposure may be more relevant in patients aged 40 or 45 years, which is the more common threshold in previous reports. Given the religious and cultural background of the study participants, alcohol consumption is considered a negligible risk factor. In addition, it is assumed that tongue cancer in the younger population can have a more infectious aspect. However, there are no definite data on the association between HPV and tongue cancer in the younger population [40]. Although updated statistics on smoking, betel quid, and shisha are required, the available data show that the prevalence of smoking in Egypt in 2010 was 22%, and the number of smokers in Egypt is estimated to increase by 8% each year [42,43]. Moreover, smoking prevalence is much less among females than males by up to 30%. Men are much more likely to use tobacco than women, and almost 96% of men who use tobacco do so daily. In addition, they are more likely to use manufactured cigarettes than shisha or smokeless tobacco. While a few women use tobacco (cigarettes (0.2%), shisha (0.3%), and smokeless tobacco (0.3%), all women who currently smoke shisha do so daily [44].

Nevertheless, other risk factors such as viral infections, including HPV, and possible genetic and environmental factors are under-studied in Egypt, which urges the need for further investigations in similar communities.

### Strengths and Limitations

Little research has been conducted in non-Western countries, specifically Muslim countries, which imposes a challenge when studying trends for tongue cancer. Our results can pave the way for further investigations of such populations that lack specific risk factors such as alcohol, as well as further investigations of possible risk factors such as environmental or genetic risk factors. In addition, there are no available updated statistics on tongue cancer in African countries generally and Egypt specifically. Therefore, we anticipate that further research will be built upon our results. However, there are some limitations such as a small sample size because the study was conducted in a single center and retrospectively, making it more liable to missing information or under-reported cases. In addition, crucial risk factors such as HPV and betel quid should be studied in this population. Although squamous cell carcinoma is the most common histological type of tongue cancer, which was mainly investigated in this study, other histological types should be evaluated.

## 5. Conclusions

The global epidemiological trends in tongue cancer have shifted toward more incidences in younger and female groups. Our study confirms the same findings regarding the younger age group. However, in our locality, no significant changes have been noted regarding gender. More multi-institutional studies are recommended to explain such changes and highlight the possible risk factors so that appropriate corrective actions can be taken into consideration.

## Figures and Tables

**Figure 1 cancers-15-05680-f001:**
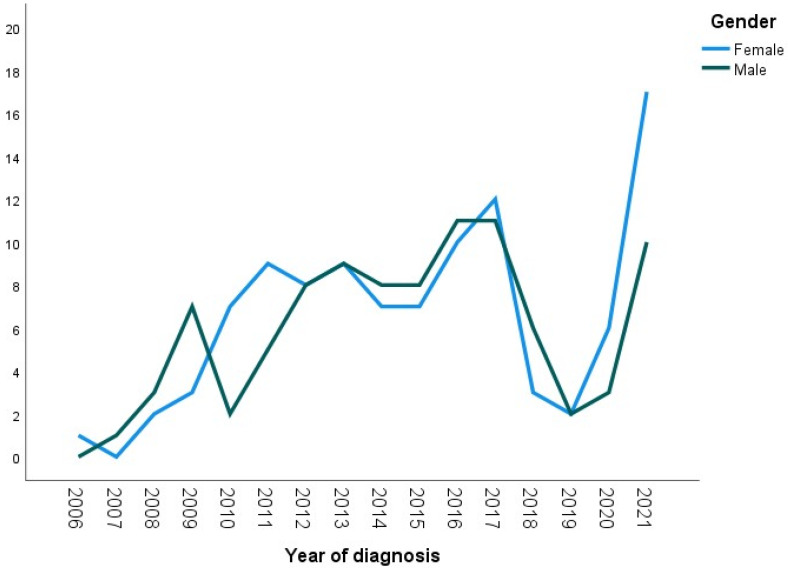
Line graph showing numbers of diagnosed males and females over the period (2006–2021).

**Table 1 cancers-15-05680-t001:** Patient characteristics for tongue cancer diagnosed between 2006 and 2021.

2006–2013			2014–2021		*p*-Value
	N	%	N	%	
Gender					0.927
Females	39	52.7%	64	52%	
Males	35	47.3%	59	48%	
Age					0.018
20–39	5	6.8%	17	13.8%	0.039
40–49	22	29.7%	53	43.1%	0.912
≥60	47	63.5%	53	43.1%	0.011
Age					0.127
<40	5	6.8%	17	13.8%	
≥40	69	93.2%	106	86.2%	
Smoking status					0.973
Smoker	17	23%	28	22.8%	
Non-Smoker	57	77%	95	77.2%	
Family history of cancer					0.002
Positive	1	1.5%	19	16.8%	
Negative	64	98.5%	94	83.2%	
Type of cancer reported in the family History					
Head and neck	0		7		
Gynecological	1		8		
GIT	0		7		
Others	0		3		

**Table 2 cancers-15-05680-t002:** Tongue cancer characteristics at presentation.

		2006–2013		2014–2021		*p*-Value
		N	%	N	%	
Site	Lateral	57	77%	103	83.7%	0.253
	Dorsum	11	14.6%	17	13.8%	
	Middle/Posterior	5	6.8%	3	2.4%	
Side	Right	36	59%	61	52.6%	0.414
	Left	25	41%	55	47.4%	
Nodal status	Positive	46	62.2%	41	33.3%	<0.001
	Negative	26	35.1%	55	44.7%	
Clinical stage	1	10	14.3%	22	26.5%	0.148
	2	13	18.6%	16	19.3%	
	3	5	7.1%	9	10.8%	
	4	42	60%	36	43.4%	
Grade	1	36	49.3%	37	45.1%	0.621
	2	30	41.1%	33	40.2%	
	3	7	9.6%	12	14.6%	

## Data Availability

All the clinical, radiological, and pathological data used in this manuscript are available on the Mansoura University medical system (Ibn Sina Hospital Management System) http://srv137.mans.edu.eg/mus/newSystem/ (accessed on 24 March 2022).

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
