# Peer review of "Shifting Epidemiology Trends in Tongue Cancer: A Retrospective Cohort Study"

_cancers, 2023, doi:10.3390/cancers15235680_

Round 1
Reviewer 1 Report
Comments and Suggestions for Authors
As you have stated already yourselves the number of patients is rather low for such an epidemiological paper. This would be the classical thesis to be answered by a multicenter study.
Could the supposedly low intake of alcohol in a Muslim country be a factor for the almost equal distribution of tongue cancer in both genders ? That means that men have a lower rate than in alcohol consuming countries .
There is no direct control to this study in another cultural background only historical comparisons.
Please comment on that .
Author Response
Dear reviewer,
Thanks very much for your valuable comment
we added the following section and references to the discussion.
"Different habits between countries may play a role in gender distribution of tongue cancer, for example, alcohol which is a worldwide known risk factor for tongue cancer. Alcohol consumption is less in Arab Muslim countries due to cultural and religious background [1]. This low intake of alcohol may show a statistical difference in gender distribution compared to non-Muslim countries. Unfortunately, we lack studies to support such a hypothesis. In addition, it is crucial to mention that drinking alcohol is less admitted by individuals in Muslim countries, especially by women[2]. However, we can't admit that men have lower rates than in the West because alcohol is not the only risk factor. Overly we need more studies, especially in Muslim communities."
[1] L. Ghandour et al., “Alcohol consumption in the Arab region: What do we know, why does it matter, and what are the policy implications for youth harm reduction?,” Int. J. Drug Policy, vol. 28, pp. 10–33, Feb. 2016, doi: 10.1016/J.DRUGPO.2015.09.013.
[2] F. Alhashimi, O. Khabour, K. Alzoubi, and S. Al-shatnawi, “Attitudes and beliefs related to reporting alcohol consumption in research studies: a case from Jordan,” Pragmatic Obs. Res., vol. Volume 9, pp. 55–61, 2018, doi: 10.2147/por.s172613.
Reviewer 2 Report
Comments and Suggestions for Authors
An interesting paper that adds to the global view of tongue cancer. It is becoming clear that there are many factors influencing the global picture.
A couple of things
The English is generally good but be careful not to omit "the" at the start of a sentence (eg abstract - although you are quite right to omit in Summary)
Table 1 might need the headings repeated if split between page 3 and 4. This might not be necessary depending on the final format.
Figure 2 would it not be better to arrange age groups in ascending or descending order?
P2 line 62 please expand MENA the first time it is used
In table 2, changes in nodal status were significant - how was this assessment made? Has there been a move to ultrasound, or is this all based on postoperative histology?
Finally, it would be helpful to have a simple graph of numbers of the total population in the three age groups at the 2 time points
Comments on the Quality of English Language
Generally good - see comments above on definite article
Author Response
Dear reviewer,
Thanks very much for your valuable comments:
- "The English is generally good but be careful not to omit "the" at the start of a sentence (eg abstract - although you are quite right to omit in Summary)"
Thorough grammar revision was performed
- "Table 1 might need the headings repeated if split between page 3 and 4. This might not be necessary depending on the final format."
The final format of Table 1 is on a separate page, page 4.
- "Figure 2 would it not be better to arrange age groups in ascending or descending order?"
Figure 2 is arranged in Ascending order now.
- "P2 line 62 please expand MENA the first time it is used"
Correction done. MENA region stands for the Middle East and North Africa region.
- "In table 2, changes in nodal status were significant - how was this assessment made? Has there been a move to ultrasound, or is this all based on postoperative histology?"
In Table 2 Nodal status assessment is based on post-operative histology.
"Finally, it would be helpful to have a simple graph of numbers of the total"
Figure 4 was added. It represents a simple graph of numbers of the total population in three age groups at the 2 times points.
Reviewer 3 Report
Comments and Suggestions for Authors
Recent observations have indicated a notable alteration in the global epidemiological patterns of tongue cancer, with an increasing prevalence among younger individuals and females. The present investigation has elucidated the emergence of this trend within the younger demographic, yet it remained inconspicuous within the female cohort within their local database from Oncology Center Mansoura University (OCMU) in Egypt. In light of these findings, the authors recommended to undertake extensive inquiries within larger population cohorts to elucidate the underlying mechanisms driving these shifts. Such research endeavors will benefit further studies reflecting trends specifically in Muslim countries and facilitate the formulation of suitable corrective measures.
Overall, this manuscript is well structured and clearly written. Following are suggestions to strengthen the impact of the manuscript:
Minor:
1. Line 94: Format issue of “Exclusion criteria”
Author Response
Dear reviewer,
Thanks very much for your nice words
The requested format issue was modified. Thanks
Round 2
Reviewer 1 Report
Comments and Suggestions for Authors
Thank you for your additions to the text.